# RETRACTED: Anticancer Activity of 2-*O*-Caffeoyl Alphitolic Acid Extracted from the Lichen, *Usnea barbata* 2017-KL-10

**DOI:** 10.3390/molecules26133937

**Published:** 2021-06-28

**Authors:** Hae-Jung Chae, Geum-Jin Kim, Barsha Deshar, Hyun-Jin Kim, Min-Ji Shin, Hyukbean Kwon, Ui-Joung Youn, Joo-Won Nam, Sung-Hak Kim, Hyukjae Choi, Sung-Suk Suh

**Affiliations:** 1Department of Bioscience, Mokpo National University, Muan-gun 58554, Jeonnam, Republic of Korea; 0o_hj_4004@naver.com (H.-J.C.); barsha.deshar.97@gmail.com (B.D.); minji4099@naver.com (M.-J.S.); 2College of Pharmacy, Yeungnam University, Gyeongsan-si 38541, Gyeongsangbukdo, Republic of Korea; canta87@ynu.ac.kr (G.-J.K.); zero9602@gmail.com (H.K.); jwnam@yu.ac.kr (J.-W.N.); 3Research Institute of Cell Culture, Yeungnam University, Gyeongsan-si 38541, Gyeongsangbukdo, Republic of Korea; 4Department of Animal Science, College of Agriculture and Life Sciences, Chonnam National University, Gwangju 61186, Republic of Korea; redallert0@gmail.com (H.-J.K.); sunghakkim@jnu.ac.kr (S.-H.K.); 5Department of Biomedicine, Health & Life Convergence Science, BK21 Four, Mokpo National University, Muan-gun 58554, Jeonnam, Republic of Korea; 6Division of Life Sciences, Korean Polar Research Institute (KOPRI), Incheon 21990, Republic of Korea; ujyoun@kopri.re.kr; 7Department of Polar Sciences, University of Science and Technology, Incheon 21990, Republic of Korea

**Keywords:** colorectal cancer, 2-*O*-caffeoyl alphitolic acid, lichen, *Usnea barbata*, apoptosis, HCT116

## Abstract

Colorectal cancer is one of the life-threatening ailments causing high mortality and morbidity worldwide. Despite the innovation in medical genetics, the prognosis for metastatic colorectal cancer in patients remains unsatisfactory. Recently, lichens have attracted the attention of researchers in the search for targets to fight against cancer. Lichens are considered mines of thousands of metabolites. Researchers have reported that lichen-derived metabolites demonstrated biological effects, such as anticancer, antiviral, anti-inflammatory, antibacterial, analgesic, antipyretic, antiproliferative, and cytotoxic, on various cell lines. However, the exploration of the biological activities of lichens’ metabolites is limited. Thus, the main objective of our study was to evaluate the anticancer effect of secondary metabolites isolated from lichen (*Usnea barbata* 2017-KL-10) on the human colorectal cancer cell line HCT116. In this study, 2OCAA exhibited concentration-dependent anticancer activities by suppressing antiapoptotic genes, such as *MCL-1,* and inducing apoptotic genes, such as *BAX*, *TP53*, and *CDKN1A*(p21). Moreover, 2OCAA inhibited the migration and invasion of colorectal cancer cells in a concentration-dependent manner. Taken together, these data suggest that 2OCAA is a better therapeutic candidate for colorectal cancer.

## 1. Introduction

Colorectal cancer is one of the life-threatening ailments leading to high mortality and morbidity worldwide. It was assumed that there were over 1.9 million new cases of colorectal cancer (including anal) and 935,000 deaths in 2020 [1]. Most cases of colorectal cancer are due to old age and lifestyle [2]. Underlying genetic alterations are also a minor cause of colorectal cancer [3]. Emerging advanced therapies such as chemotherapy, surgical excision, and radiotherapy remain the gold standard for combating colorectal cancer. However, the prognosis for malignant colorectal patients remains unsatisfactory. To achieve a better treatment for colorectal cancer, we should develop a thorough concept of the mechanism of programmed cell death better known as apoptosis. For instance, a defect in apoptosis in cancer cells is associated with resistance to therapy [4]. Additionally, in cancer, inactivation of the death promoter Bax stimulates apoptotic resistance, a mutation in the p53 gene, the expression of antiapoptotic proteins such as Bcl-2, and so on. Furthermore, a defect in apoptosis promotes tumor invasion and metastasis [5]. Thus, targeting apoptotic and antiapoptotic genes is proposed as a better therapeutic candidate for colorectal cancer.

Recently, natural products have attracted the attention of researchers in the search for targets to fight against cancer. They are considered vital to the discovery of cancer treatments [6]. An example of these natural products is lichen, which is a symbiotic association of fungi (mycobionts) and algae (phytobionts). They are ecologically diverse and are distributed from the tropics to the polar regions. In particular, they have adapted well to extreme environments, such as the polar and alpine regions, deserts, and volcanic regions. Lichens also contain various secondary metabolites exhibiting excellent physiological activities, including basic antioxidant active substances that protect cells from various environmental stresses. Until now, plants have been widely used as biological resources for the development of new natural drug candidates. In addition, the research and development of various higher fungi, including mushrooms, have progressed among microorganisms. However, lichens have high biodiversity and account for more than one-third of all fungi. However, the mechanism studies of natural products extracted from lichens are still in their infancy. Recent studies demonstrated that lichen-derived metabolites with aliphatic, cycloaliphatic, aromatic, and terpenic structures have demonstrated anticancer [7], antiviral [8], anti-inflammatory [9], antibacterial [10], analgesic, antipyretic [11], antiproliferative [12,13], and cytotoxic effects [14] on various cell lines [15,16]. Therefore, it is estimated that there are about 18,500 species of lichens with more than a thousand metabolites [17]. However, exploration of the biological activities of lichens’ metabolites is limited.

As part of our efforts to search for bioactive compounds from lichen, *Usnea barbata* was selected for further studies. The genus *Usnea* was used in traditional medicine and dyeing for thousands of years [18]. Recently, *Usnea* sp. has been used as a source of usnic acid in the cosmetic industry. In pharmacology, *U. barbata* was reported to exhibit bioactivities, such as antioxidant, antimicrobial [19], and anti-inflammatory [20]. In particular, extracts of *U. barbata* were previously reported to have anticancer properties against several cancer cell lines, including oral cancers (Ca9-22, OECM-1, CAL 27, HSC3, and SCC9) [21,22], melanoma (FermX), and colorectal carcinoma (LS174) [19]. Usnic acid, which is a well-known major compound of *U. barbata*, was found to be an anticancer compound against melanoma (FermX), colorectal carcinoma (LS174) [19], and human lymphocytes (V79 and A549) [23]. However, the bioactive metabolites of *U. barbata* with cytotoxicity against the human colorectal cell line (HCT116) have not been demonstrated. In this study, the chemical investigations of the extract of *U. barbata* 2017-KL-10, followed by bioactivity evaluations, led to the isolation of eight compounds (**1–8**) (three terpenoids, four depsides, and a dibenzofurans) and the identification of cytotoxic compounds against HCT116. Among the tested compounds, 2-*O*-caffeoyl alphitolic acid (**1**, 2OCAA) demonstrated cytotoxic activities in the human colorectal cell line (HCT116). Thus, we examined the tumor-suppressive effects of compound **1** on the basis of its molecular mechanisms on cell growth, apoptosis, cell migration, and invasion processes in HCT116 cells.

## 2. Results and Discussion

### 2.1. Structure Identification of Compounds ***1***–***8***

Compound **1** was isolated as a colorless oil. The low resolution-fast atom bombardment-mass spectrometry (LR-FAB-MS) analysis of **1** revealed a proton adduct ion at m/z 635 and a sodium adduct ion at *m*/*z* 657. The ^1^H nuclear magnetic resonance (NMR) spectrum of **1** also demonstrated six methyl singlet signals (δ_H_ 1.68, 1.04, 1.02, 1.01, 0.98, and 0.86 ppm), an oxygenated signal (δ_H_ 5.05 ppm), two sets of olefinic signals (δ_H_ 7.56/6.28 and 4.70/4.57 ppm), and a series of proton signals of an ABX aromatic ring (δ_H_ 7.04, 6.94, and 6.77 ppm). Through a careful inspection of the ^13^C NMR and Heteronuclear Multiple Bond Correlation (HMBC) spectra, 39 carbon signals were identified, including two carbonyl carbon signals (δ_C_ 180.2 and 169.3 ppm). Similarly, on the basis of 1D and 2D NMR data analysis, compound **1** was identified as 2-*O*-caffeoyl alphitolic acid by comparison with previously reported 1D NMR data (Appendix A) [24].

Compound **2** showed an identical proton adduct ion and sodium adduct ion at m/z 635 and 657, respectively, in LR-FAB-MS. However, the ^1^H NMR spectrum of **2** was similar to that of **1**, thus supporting a caffeoyl group attached to a triterpene with a closely related skeleton. Similarly, the structure of **2** was elucidated as 2-*O*-caffeoyl maslinic acid through the 1D NMR data comparison with those in the literature (Appendix A) [25].

In addition, from the LR-FAB-MS data, compound **3** exhibited a protonated ion at *m*/*z* 499. The characteristic proton signals, including five methyl singlets (δ_H_ 1.72, 1.69, 0.97, 0.95, and 0.82 ppm), two oxygenated methyl signals (δ_H_ 3.67 and 3.65), and four olefinic proton signals (δ_H_ 4.84, 4.74, 4.64, and 4.61 ppm). The ^1^H NMR spectrum also demonstrated two isopropenyl and two methyl ester groups. On the basis of the 1D and 2D NMR data analysis, compound **3** was identified to be 3,4-seco-lupa-4(23),20(29)-diene-3,28-dioic acid, dimethyl ester by comparing NMR data with the report (Appendix A) [26].

In addition, the structures of **4**–**8** were identified as propylresorcinol (**4**) [27], divaric acid (**5**) [28], divaricatic acid (**6**) [29], divaricatinic acid methyl ester (**7**) [30], and (+)-usnic acid (**8**) [31], respectively, by comparing the experimental and reported spectroscopic data (Figure 1). Compounds **1–3** were especially isolated from the lichen for the first time.

### 2.2. 2-O-Caffeoyl Alphitolic Acid Inhibits HCT116 Cell Growth

The cytotoxicity of **1**–**8** in HCT116 cells was tested (data not shown), and among the tested isolates, compound **1**, 2-*O*-caffeoyl alphitolic acid (2OCAA), demonstrated the most potent cytotoxicity. Compound **1** was previously reported from the barks of *Daphniphyllum oldhami* [32] and *Alnus viridis* ssp. *viridis*. In a previous study, the cytotoxicity of **1** was evaluated in human melanoma (A375) and lung carcinoma (A549) cell lines. Compound **1** also demonstrated an anticancer activity toward the A375 cell line (IC_50_ 6.3 ± 0.3 µM; cisplatin 4.1 ± 0.2 µM) along with moderate selectivity (1.25). Compound **1** also reported inhibitory activities of topoisomerase in an in silico study. In this study, the molecular mechanism of cytotoxicity of 1 on HCT116 cells was investigated. Compound **1** is a pentacyclic lupane-type triterpene ester with a caffeoyl group, as presented in Figure 2A. Typical lupane-type triterpenoids, such as betulin, betulinic acid, and lupeol, were reported to exhibit anticancer activities in various cancer cell lines, and natural products with lupane skeleton are known to have antitumor potentials [33]. To determine the cytotoxic effect of **1** on colorectal cancer cells, we conducted a cellular proliferation analysis using 3-(4,5-dimethylthiazol-2-yl)-5-(3-carboxymethoxyphenyl)-2-(4-sulfophenyl)-2H-tetrazolium (MTS) assay on HCT116 cells. The cells were treated with different concentrations of 2OCAA (0, 6.25, 12.5, 25, and 50 μM) and incubated for 24 h. The proliferation of HCT116 cells was gradually but significantly suppressed by 12.5–50 μM, but not 6.25 μM, of 2OCAA in a concentration-dependent manner. The highest concentration (50 μM) exhibited a stronger inhibitory effect than the lower concentrations (12.5 or 25 μM) (Figure 2B). This result is consistent with previous studies showing that triterpenoids have cytotoxic properties against various tumor cells such as melanoma, liver cancer, breast cancer, and glioma [34,35]. Subsequently, we observed the morphological features of the 2OCAA-exposed cells under the microscope. Cells dying by apoptosis exhibit common characteristic morphological changes, such as shrinkage and fragmentation into the membrane-bound bodies [36]. In this study, the 2OCAA-exposed cells demonstrated morphological hallmarks of apoptosis, suggesting that HCT116 cells are induced to undergo apoptosis upon treatment with 2OCAA (Figure 2C). These results suggest that 2OCAA exhibits anticancer activities in a concentration-dependent manner, including the inhibition of cellular proliferation and induction of apoptosis against colorectal cancer cells (HCT116).

### 2.3. 2-O-Caffeoyl Alphitolic Acid Enhances Cell Apoptosis

Apoptosis or programmed cell death is the natural death of a cell and is important for embryogenesis, organ involution, maintenance of homeostasis, and biodefense systems of the body [37]. In cells undergoing apoptosis, an intracellular signaling pathway operates cells autonomously, which leads to the death and disposal of the cells through the modulation of apoptosis-associated genes. As a result of the morphological features of apoptosis in the 2OCAA-exposed cells (Figure 2C), we conducted flow cytometry to evaluate the apoptotic effect of 2OCAA on the HCT116 cells. In response to the treatment with 2OCAA, consistent apoptosis was observed in these cells. Similarly, we observed a significant concentration-dependent increase in apoptotic cells after the treatment with 2OCAA (Figure 3), inducing apoptosis at 12.92, 22.99, 27.76, and 60.45% levels, respectively, in response to the exposure of 6.25, 12.5, 25, and 50 μM concentrations compared with 11.72% in the control group. To further investigate the molecular mechanisms underlying these interesting results, we conducted real-time PCR to observe the expression levels of apoptotic marker genes, such as Mcl-1, Bax, p53 and p21 in the HCT116 cell line. The expression levels of antiapoptotic gene Mcl-1 were remarkably decreased in a concentration-dependent manner, whereas the apoptosis-associated genes, Bax, p53 and p21, were augmented consistently as indicated by the flow cytometry data (Figure 4). This is consistent with literature data showing that typical lupane-type triterpenoids including botulin, betulinic acid, and lupeol can induce apoptosis in various human cancers such as lung cancer, hepatic cancer, gastric cancer and breast cancer [33]. In particular, it is remarkable that lupine-type triterpenoids have shown interesting effects against colorectal cancer. For example, betulinic acid and its derivatives exert potent antitumor activities by inhibiting tumor growth through downregulation of VEGF (vascular endothelial growth factor), which plays a crucial role in regulating angiogenesis [38] and triggering the extrinsic pathway of apoptosis [39]. It can also modulate the expression levels of different Bcl-2 family proteins [33]. In addition, the treatment of betulinic acid resulted in upregulation of the proapoptotic protein Bax in neuroblastoma, glioblastoma, and melanoma cells. Taken together, our data suggest that 2OCAA, one of the lupane-type triterpenes, induces apoptosis in cancer cells through the modulation of apoptosis-associated genes and has important biological potential in the treatment for colorectal cancer.

### 2.4. 2-O-Caffeoyl Alphitolic Acid Inhibits Cell Migration and Invasion

Migration and invasion into the surrounding tissue, cell blood, and lymphatic vessels are known as the initial steps of metastasis [40]. Cell migration and invasion are associated with the production of reactive oxygen species (ROS), development of chemo-resistant cancer stem cells, introduction of mutations in DNA damage repair genes, and contribution of microRNAs [41]. To study the role of 2OCAA in cell migration and invasion, we conducted a wound-healing assay or scratch assay. Wound healing is a complex and fragile procedure that has inflammatory, proliferative, and remodeling phases. In particular, cancer cell migration is essential for wound healing. The wound-healing effects of 2OCAA at concentrations without cytotoxic doses (3.125 and 6.25 μM) were evaluated in HCT116 cells using the MTS cell viability assay (Figure 2B). As shown in Figure 5, 2OCAA significantly inhibited wound healing in HCT116 cells in a concentration-dependent manner. This suggests that 2OCAA is associated with the suppression of cell migration. In particular, a concentration of 2OCAA (6.25 μM) completely inhibited the migration of HCT116 cells, showing a similar pattern to that of cells in the control group. Next, the inhibitory capacities of 2OCAA on cancer cell migration and invasion were investigated using the Boyden chamber system. As presented in Figure 6, both the migration and invasive capacities of HCT116 cells were significantly inhibited by 2OCAA in a concentration-dependent manner. These data suggest that 2OCAA may regulate metastasis by effectively suppressing cancer migration and invasion.

Recently, there has been a growing interest in natural products as therapeutic agents worldwide. Among them, lupane-type triterpenes such as betulin, betulinic acid, and lupeol have shown multiple biological activities against various cancer cell lines and encouraging antitumor effects [33,34,35,36,37,38,39]. In this study, 2OCAA showed concentration-dependent anticancer activities such as cytotoxicity, induction of apoptosis, inhibition of cell migration and invasion of colorectal cancer. Further in vitro and in vivo studies and human clinical trials are needed to elucidate their mechanism and therapeutic potential. However, there is great promise for this class of compounds as therapeutic agents to treat malignancy.

## 3. Materials and Methods

### 3.1. General Experimental Procedures

The ^1^H and ^13^C NMR spectra were recorded at 25 °C using a 600 MHz Fourier transform nuclear magnetic resonance device (VNS600; Agilent Technologies, Santa Clara, CA, USA) at the Core Research Support Center for Natural Products and Medical Materials (CRCNM) in the Yeungnam University. Medium pressure liquid chromatography (MPLC) was also conducted using Biotage Isolera (Biotage, Uppsala, Sweden). In addition, high-performance liquid chromatography (HPLC) was conducted on a Waters system (Waters 1525 pump and Waters 996 PDA; Waters, Milford, MA, USA) with a semi-preparative HPLC column (Phenomenex Luna C18(2), 10 mm × 250 mm, 5 μ (Phenomenex, Torrance, CA, USA) and an analytical HPLC column (Phenomenex Luna C18(2), 4.6 mm × 250 mm, 5 μ (Phenomenex, Torrance, CA, USA).

### 3.2. Lichen Material

The lichen was collected in November 2017 from San Juan, Chile, by Dr. U. J. Youn, Department of Life Sciences, Korea Polar Research Institute. Dr. Youn identified the lichen as *Usnea barbata*. The voucher specimen (2017-KL-10) was stored in the College of Pharmacy, Yeungnam University, Korea.

### 3.3. Extraction and Isolation

A portion of the lichen (237.7 g) was extracted using 50% methylene chloride (MC) in methanol (MeOH) (4 × 300 mL) at room temperature. Then, the residue was evaporated under reduced pressure to obtain a crude extract (23.4 g). The extract was suspended in distilled water (100 mL) and partitioned three times with an equal volume of MC. Then, MC soluble layers were combined and dried completely. Next, the dried MC soluble layers were subjected to liquid–liquid extraction with hexanes and MeOH to obtain the hexane soluble (4.1 g) and MeOH (10.1 g) fractions. The MeOH fraction was further fractionated into MeOH soluble (7.5 g) and insoluble (2.6 g) fractions based on the MeOH solubility. The MeOH insoluble fraction was identified as compound **8**. The aqueous layer was then extracted with ethyl acetate and n-butanol (BuOH) to obtain ethyl acetate (1.6 g), n-BuOH (4.9 g), and H_2_O (5.5 g) fractions. The MeOH soluble fraction was also subjected to further fractionation on a SNAP cartridge KP-SIL 100 g (Biotage, Uppsala, Sweden) using normal-phase MPLC (Biotage, Uppsala, Sweden), with a gradient condition with MC and MeOH to yield eight sub-fractions (Fr. A–H).

Fraction A (202.4 mg) was subjected to HPLC in 80% can and distilled water conditions on the preparative column (Hector M C18 20 × 250 mm) to obtain seven sub-fractions, including compound **6** (26.8 mg, Rt = 29.4 min). Reversed-phase HPLC was employed to further purify fraction A-7 (22.6 mg) under the following conditions: Phenomenex Luna C18(2) 10 × 250 mm, 5 μ (Phenomenex, Torrance, CA, USA), 2.5 mL/min, MeOH/H_2_O = 83:17 (42 min)→100:0 (47 min)→100:0 (70 min). Subsequently, the collected peak in fraction A-7 (Fr. A-7-L) was identified as compound **3** (5.3 mg, Rt = 69.2 min). Similarly, fraction E (877.3 mg) was further fractionated on a SNAP cartridge KP-SIL 100 g (Biotage, Uppsala, Sweden) manually packed with C18 gels (Merck KGaA, Darmstadt, Germany) via reversed-phase MPLC (Biotage, Uppsala, Sweden) with a gradient elution with MeOH and H2O to obtain 17 fractions (Fr. E-1–E-17). Moreover, compounds **5** (5.9 mg, Rt = 5.5 min) and **4** (14.1 mg, Rt = 18.3 min) were purified from fraction E-3 (42.1 mg) via reversed-phase HPLC. A gradient elution of CH_3_CN and H_2_O [ACN/H2O = 40:60 (15 min) → 60:40 (60 min), as well as a Phenomenex Luna C18(2) 10 × 250 mm, 5 μ, 2 mL/min] was used. Furthermore, the MeOH insoluble part of fraction E-10 (14.3 mg) was confirmed as compound **7**.

Fraction E-11 (61.7 mg) was chromatographed on an RP-HPLC (Phenomenex Luna C18(2) 10 × 250 mm, 5 μ (Phenomenex, Torrance, CA, USA), 3 mL/min, MeOH/H_2_O = 85:15 isocratic mode), which yielded seven sub-fractions (Fr. E-11-A–E-11-G). Reversed-phase HPLC was employed to further purify fraction E-11-C (9.1 mg) and E-11-D (6.4 mg) on an analytical column (Phenomenex Luna C18(2) 4.6 × 250 mm, 5 μ (Phenomenex, Torrance, CA, USA)) with a flow rate of 1 mL/min. A solvent mixture of 70 and 72% acetonitrile in H_2_O was used in an isocratic condition, respectively, to obtain compounds **1** (3.5 mg, Rt = 17.6 min) and **2** (3.7 mg, Rt = 28.3 min) as a single peak.

### 3.4. Cell Culture and Cell Proliferation Assay

The human cancer cell line (HCT116) in this study was purchased from the Korean Cell Line Bank (Seoul, Korea) and cultured at 37 °C and 5% CO_2_ in Dulbecco’s Modified Eagle’s Medium (DMEM). Then, 10% fetal bovine serum and 1% penicillin–streptomycin were mixed with DMEM. Cell viability was determined via the MTS assay following the manufacturer’s protocol (Promega). In brief, the cells were subcultured in 96-well plates and starved for 12 h in culture media with 0.1% FBS, and 2OCAA was treated as described in the Results section and illustrated in the figure legends. Subsequently, 10 μL of MTS solution was added to the plate and incubated in a cell culture incubator for 20–60 min, followed by measuring observance at 490 nm. Also, the absorbance was measured using a Microreader (Molecular Device, Sunnyvale, CA, USA), and the experiment was repeated three times. The mean of the obtained values was used for the reporting of the result.

### 3.5. Apoptotic Analysis

The cells treated with 2OCAA were prepared for staining using an Apoptosis Detection Kit (Thermo Fisher Scientific, Waltham, MA, USA). The desired cells were then stained using Annexin V conjugated with Fluorescein (FITC) and propidium iodide with Phycoerythrin (PE) as recommended by the manufacturer’s protocol. Subsequently, FACSCalibur (Becton Dickinson, Franklin Lakes, NJ, USA) was used to determine cell death with 1 × 10^5^ cells. Overall, the ratio of cells with positive signals in each panel is summarized in the figure.

### 3.6. RT-PCR for Apoptosis-Associated Genes

Total RNA was extracted using TRIzol Reagent Invitrogen (Cat# 15596–018) following the manufacturer’s protocols. Specifically, the pellet obtained from 5 × 10^6^ cells was lysed using 1 mL of TRIzol solution. At the end of the extraction, the isolated RNA was dissolved in 40 µL of RNase-free water and incubated for 10 min at 55 °C. Next, 1 μg of total RNA was synthesized from its cDNA using a High-Capacity cDNA Reverse Transcription Kit (Thermo Fisher Scientific). RT-PCR was also performed by monitoring the increase in the amount of SYBR-Green in real time using a CFX Connect Real-Time System (Bio-Rad, Hercules, CA, USA). The real-time PCR conditions were as follows: 3 min initial denaturation at 94 °C and 35 cycles of 30 s at 94 °C (denaturation), 15 s at 57 °C (annealing), 30 s at 72 °C (extension). The primer sequences used are presented in Appendix A.

### 3.7. Wound-Healing Assay

The indicated cells were seeded in a six-well plate and cultured until confluent. The wound was also induced by gently making a straight scratch using a (yellow) pipette tip. After 24 h, images of migrated cells in the wound area were taken using a digital camera. All scratch assays were performed three times.

### 3.8. Migration and Invasion Assay

The overall experimental procedures were performed following the manufacturer’s protocol (Calbiochem). The cell suspension was briefly placed in an upper chamber and then in a reduced serum medium (1% FBS) for 24 h. Subsequently, the cells were treated with 2OCAA. Migratory cells passed through a polycarbonate membrane and attached to the bottom side. In contrast, non-migratory cells stayed in the upper chamber. In addition, complete media containing 10% FBS were added to the lower chamber, which served as a chemoattractant. After 24–48 h, the migrated and invasive cells were stained with a Cell Stain Solution (400 μL) and quantified.

### 3.9. Statistical Analysis

The data were analyzed by one-way ANOVA with Tukey’s post hoc test as indicated in the figure legends. Data are shown as mean  ± SD. *p*-values smaller than 0.05 were considered as statistically significant (* *p*  <  0.05, ** *p*  <  0.01). All statistical tests were performed in GraphPad Prism 7.01 (GraphPad Software, San Diego, CA, USA). 

## 4. Conclusions

In this study, we demonstrated that 2OCAA, a secondary metabolite from *Usnea barbata* 2017-KL-10, a Chilean lichen, exhibited potential anticancer activities on HCT116 cell lines. The anticancer activities included effective inhibition of cell growth, induction of apoptosis through the modulation of apoptosis-associated genes, and suppression of migration and invasion. To the best of our knowledge, this study is the first to analyze the role of 2OCAA in the colorectal cancer cell line (HCT116). On the basis of the results from these studies, more specific functions of the anticancer activities of 2OCAA in other cell lines or in vivo are required for clinical trials. Taken together, our findings indicate that 2OCAA is a potential therapeutic candidate for inhibiting colorectal cancer.

## Figures and Tables

**Figure 1 molecules-26-03937-f001:** The structures of compounds **1**–**8**.

**Figure 2 molecules-26-03937-f002:** 2OCAA inhibits cellular growth in a concentration-dependent manner. (**A**) Chemical structure of 2OCAA, (**B**) MTS assay using HCT116 cell lines, (**C**) Microscopic images of HCT116 cells treated with different concentrations of 2OCAA (200× magnification. Bars represent ± SEM, and *p*-values were calculated by one-way ANOVA with Tukey’s post hoc test (*n* = 3, * *p* < 0.05, ** *p* < 0.01).

**Figure 3 molecules-26-03937-f003:** 2-O-Caffeoyl alphitolic acid-mediated apoptosis in HCT116 cancer cells treated with different concentrations of 2OCAA. 2OCAA-mediated apoptosis significantly increased the proportion of apoptotic cells relative to that in the untreated control cells. The quadruplicate plot represents early apoptosis (annexin-V+ and PI−) and late apoptosis or necrosis (annexin-V+/− and PI+) stages.

**Figure 4 molecules-26-03937-f004:** Expression of apoptotic and antiapoptotic genes involved in the HCT116 cell line. 2OCAA significantly suppressed the antiapoptotic gene Mcl-1 and induced apoptotic genes, such as Bax, p53, and p21. Bars represent ± SEM, and *p*-values were calculated by one-way ANOVA with Tukey’s post hoc test (*n* = 3, * *p* < 0.05, ** *p* < 0.01).

**Figure 5 molecules-26-03937-f005:** Effect of 2OCAA on wound healing in HCT116 cells. 2OCAA significantly suppressed the wound-healing ability of HCT116 cells in a concentration-dependent manner relative to the untreated control cells. The gap-filling capacity was determined by measuring the distance between the gaps using the ImageJ software. The overall effect is summarized in the bottom panel of each figure. Bars represent ± SEM, and *p*-values were calculated by one-way ANOVA with Tukey’s post hoc test (*n* = 3, * *p* < 0.05, ** *p* < 0.01).

**Figure 6 molecules-26-03937-f006:** Anti-invasive and anti-migratory activity of 2OCAA in HCT116 cells. 2OCAA significantly suppressed the invasion and migration of HCT116 colorectal cancers in a concentration-dependent manner relative to the control. Normal medium containing 10% fetal bovine serum was added as a chemoattractant. After 24 h, migratory and invasive cells were monitored. The data on the right side of each figure are summarized. Bars represent ±SEM, and *p*-values were calculated by one-way ANOVA with Tukey’s post hoc test (*n* = 3, * *p* < 0.05, ** *p* < 0.01).

## Data Availability

Data contained within the article or Appendix A.

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
