# Peer review of "Anticancer Activity of 2-O-caffeoyl Alphitolic Acid Extracted from the Lichen, Usnea barbata 2017-KL-10"

_molecules, 2021, doi:10.3390/molecules26133937_

Round 1

Reviewer 1 Report

The article " Anticancer activity of 2-O.caffeoyl alphitoloc acid extracted from the lichen, Usnea barbata 2017-KL-10” describes the antitimoral properties of this extract. In general the article is interesting but I have some major concerns:

  • Discussion of results is really poor. Authors decide to include Discussion in Results section but they have not compare their results with other previous publications of other groups, with this or similar extracts.
  • The quality of Fig2c images is bad, Image quality should be improved . Magnification information should be also included.
  • Please review the asterisks of statistical analysis of fig 4. There is only one asterisk in Mcl-2 expression study of cells treated with 50uM dose. It does not fit qith the two asterisks of 25 uM. Similar happens in Bax mRNA levels (with 50uM) where there are 2 asterisks whereas there is only one in 25uM, and in p21 (with 25uM) where there is 1 asterisk whereas there are two in 12.5uM.
  • The sentence The wound- healing effects…. MTS cell viability assay (lines 189-191) is no clear
  • The sentence “These data suggest that 2OCAA…. Cancer migration and invasion” (lines 199-200) overestimate the results. 2OACAA regulates crutial steps in metastasis formation but authors have not perform metástasis or in vivo studies tos ay that this extract regulates metastastasi.

Author Response

Comments and Suggestions for Authors

The article " Anticancer activity of 2-O.caffeoyl alphitoloc acid extracted from the lichen, Usnea barbata 2017-KL-10” describes the antitimoral properties of this extract. In general the article is interesting but I have some major concerns:

Response: We appreciate the time and efforts of the reviewer in consideration of the original manuscript and believe that the comments have helped us to improve the manuscript. All the corresponding changes in response to the reviewer’s comments are marked in red.

Comment 1: Discussion of results is really poor. Authors decide to include Discussion in Results section but they have not compare their results with other previous publications of other groups, with this or similar extracts.

Response: First, we greatly appreciate your comments, critical to improve overall quality of this manuscript. As recommended from the reviewer, we provided more information about the Discussion in Result section, comparing our results with other previous publication (line 168-181).

Comment 2: The quality of Fig2c images is bad, Image quality should be improved. Magnification information should be also included.

Response: Thank you for your suggestion. Figure 2C showed microscopic images of HCT116 cell treated with 2OCAA. As the reviewer pointed out, we think that the quality of Figure 2C images is a little bad. In our laboratory, the resolution capability of our laboratory equipment, especially the microscope, is not very good. We would appreciate if you understand this technical limitation to interpret the result. However, the figure with 300dpi efficiently showed the morphological characteristics of apoptotic cells exposed to 2OCAA. Magnification information was included in figure legend (line 149).

Comment 3: Please review the asterisks of statistical analysis of fig 4. There is only one asterisk in Mcl-2 expression study of cells treated with 50uM dose. It does not fit qith the two asterisks of 25 uM. Similar happens in Bax mRNA levels (with 50uM) where there are 2 asterisks whereas there is only one in 25uM, and in p21 (with 25uM) where there is 1 asterisk whereas there are two in 12.5uM.

Response: Thank you for your suggestion. As the reviewer pointed out, we carefully reviewed the manuscript. In particular, we recognized and corrected our statistical mistakes in analyzing our experimental results. Again, we greatly appreciate your comments to correct our mistakes.

Comment 4: The sentence The wound- healing effects…. MTS cell viability assay (lines 189-191) is no clear

Response: Thank you for your suggestion. For clarity, the sentence has been rewritten as below; “The wound-healing effects of 2OCAA at concentrations without cytotoxic doses (3.125 and 6.25 μM) were evaluated in HCT116 cells using the MTS cell viability assay (Figure 2B)”

Comment 5: The sentence “These data suggest that 2OCAA…. Cancer migration and invasion” (lines 199-200) overestimate the results. 2OACAA regulates crucial steps in metastasis formation but authors have not performed metastasis or in vivo studies tos ay that this extract regulates metastaticity.

Response: Thank you for your suggestion. As the reviewer pointed out, in this experiment, a direct experiment (in vivo) that 2OCAA is involved in cancer metastasis was not performed. However, by measuring the ability of cancer cells to migrate and invade, which is the most important mechanism in the cancer metastasis process, we propose the hypothesis that 2OCAA can regulate the cancer metastasis process. For clarity, the sentence has been rewritten as below; “These data suggest that 2OCAA may regulate metastasis by effectively suppressing cancer migration and invasion.” If a sufficient amount of 2OCAA material is obtained in the future, we would like to prove a direct relationship between cancer metastasis marker gene expression and in vivo experiments. The lichen used in this study does not grow naturally in Korea and is overseas, so there is a limit to the collection, and it is difficult to conduct additional experiments due to the small amount of material currently in stock.

Reviewer 2 Report

In this paper, Chae and coll. assessed the anticancer effect of secondary metabolites isolated from lichen (Usnea barbata 2017-KL-10) on the human colorectal cancer cell line HCT116. In detail, 2OCAA showed concentration-dependent anticancer activities by suppressing anti-apoptotic genes, such as Bcl-2, and inducing apoptotic genes, such as Bax, p53, and p21. Moreover, 2OCAA inhibited the migration and invasion of colorectal cancer cells in a concentration-dependent manner. Overall, the scientific results reported may also be interesting, however, there are many inaccuracies that deserve further consideration.

Major revisions:

- One major limitation of the present study is the exclusive use of one authenticated cell line (HT116) with, for example, KRAS mutation. Additional authenticated colorectal cancer lines with a common mutation in genes as PTEN, TP53, ERBB2, and KRAS would strengthen the conclusions of the study. To this end, the authors must perform, at least in part, cytotoxic assay by using at least two different kinds of colon cancer cell lines.

- Pag. 1 Line: 23, the authors stated: 2OCAA exhibited concentration-dependent anti-cancer activities by suppressing anti-apoptotic genes, such as Bcl-2. Thereafter (pag. 6, figure 4), the authors showed in the figure the mRNA expression levels of Mcl-2. Can the authors please explain this incongruency?

- Regarding the cell proliferation assay by using the MTS assay, I have several concerns. First of all, how can the authors claim that 2OCAA affects the cell growth of HCT116 cells if they did not perform serum starvation prior to drug treatment?  Since the serum-starvation is widely used for cell synchronization during the cell cycle, I would suggest to the authors to taking into account this point, otherwise, it makes not much sense to talk to inhibition of cell proliferation/cell growth. Since the MTS is by definition a metabolic assay (also used for viability assay), I would suggest to the authors to report the metabolic % activity instead of the relative cell growth. Please revise.

- The potential for 2OCAA to prevent colon cancer will depend on its ability to act on neoplastic cells preferentially over normal colon cancer cells. To this end, I would like to ask the authors: What is the action of 2OCAA on healthy epithelial colon cells? It needs to be established that 2OCAA does not cause cell death or cell growth inhibition of normal gut epithelial cells if it could represent a therapeutic candidate for colorectal cancer (as the authors stated in line31, page 1).  The study does not address this point.

- Since the authors report the occurrence of both early and/or late apoptosis after 2OCAA, the study itself lacks true mechanistic insights as to how 2OCAA elicits anti-tumor effects in human colorectal HCT116 cancer cells. In other words, crucial regulators of apoptotic machinery (cleaved PARP, caspase-3, caspase-9, and caspase-8) should be investigated to justify the obtained results also related to the control of mitochondrial permeability (e.g induction of Bax, inhibition of BCL-2).

- The manuscript must be revised by a native English speaker as it contains a lot of grammar errors and many sentences are very difficult to be followed.

- The figures' resolution is very poor, and the figure legends should be more descriptive so that they stand alone from the text.

- The discussion needs to be rewritten in order to be more fluid and not just a list of data.

Minor revisions:

Line 288, page 10 – Please indicate the number of collected events for flow cytometry analysis.

Line 298, page 10 – Please indicate the thermal cycle protocol for qRT-PCR analysis (denaturation, annealing, extension….).

Author Response

Comments and Suggestions for Authors

In this paper, Chae and coll. assessed the anticancer effect of secondary metabolites isolated from lichen (Usnea barbata 2017-KL-10) on the human colorectal cancer cell line HCT116. In detail, 2OCAA showed concentration-dependent anticancer activities by suppressing anti-apoptotic genes, such as Bcl-2, and inducing apoptotic genes, such as Bax, p53, and p21. Moreover, 2OCAA inhibited the migration and invasion of colorectal cancer cells in a concentration-dependent manner. Overall, the scientific results reported may also be interesting, however, there are many inaccuracies that deserve further consideration:

Response: We appreciate the time and efforts of the reviewer in consideration of the original manuscript to improve the manuscript. All changes in the text (additions and deletions over the previous submission) are marked in red.

Comment 1: One major limitation of the present study is the exclusive use of one authenticated cell line (HT116) with, for example, KRAS mutation. Additional authenticated colorectal cancer lines with a common mutation in genes as PTEN, TP53, ERBB2, and KRAS would strengthen the conclusions of the study. To this end, the authors must perform, at least in part, cytotoxic assay by using at least two different kinds of colon cancer cell lines.

Response: First, we greatly appreciate your comments, critical to improve overall quality of this manuscript. As the reviewers pointed out, one of the major limitations of the current study is that the experiment used only the cell line (HT116). We also acknowledge the need for studies involving a variety of colorectal cancer cells. However, the lichen used in this experiment was collected in Chile in November 2017 (intermediate transit area to the Antarctic Research Institute) and the amount of 2OCAA extracted was insufficient for various molecular biology experiments. Also, there are currently no additional amount left to proceed with the experiment. Therefore, there is the limitation in conducting the experiment. In the future, we intend to secure additional substances and study more specific anticancer effects along with the points pointed out by the reviewer.

Comment 2: Pag. 1 Line: 23, the authors stated: 2OCAA exhibited concentration-dependent anti-cancer activities by suppressing anti-apoptotic genes, such as Bcl-2. Thereafter (pag. 6, figure 4), the authors showed in the figure the mRNA expression levels of Mcl-2. Can the authors please explain this incongruency?

Response: We greatly appreciate your comment. In the process of writing the manuscript, Mcl-1, a pro-survival member of Bcl-2 family, was confused with Bcl-2. We have corrected this mistake in the manuscript. Again, thanks for your comment.

Comment 3: Regarding the cell proliferation assay by using the MTS assay, I have several concerns. First of all, how can the authors claim that 2OCAA affects the cell growth of HCT116 cells if they did not perform serum starvation prior to drug treatment?  Since the serum-starvation is widely used for cell synchronization during the cell cycle, I would suggest to the authors to taking into account this point, otherwise, it makes not much sense to talk to inhibition of cell proliferation/cell growth. Since the MTS is by definition a metabolic assay (also used for viability assay), I would suggest to the authors to report the metabolic % activity instead of the relative cell growth. Please revise.

Response: We greatly appreciate your comment. Usually, serum starvation is performed to check the drug response more accurately. Because the serum in the culture media contains many nutrients, the exact effect of the drug to be treated by the serum may be reduced. In the present study, cells were incubated in 0.1% FBS for 12 hours prior to material treatment. This is detailed in Materials and Methods (line 288). In addition, as reviewer’s suggestion, we reported the metabolic % activity instead of the relative cell growth in the figure 2.

Comment 4: The potential for 2OCAA to prevent colon cancer will depend on its ability to act on neoplastic cells preferentially over normal colon cancer cells. To this end, I would like to ask the authors: What is the action of 2OCAA on healthy epithelial colon cells? It needs to be established that 2OCAA does not cause cell death or cell growth inhibition of normal gut epithelial cells if it could represent a therapeutic candidate for colorectal cancer (as the authors stated in line31, page 1).  The study does not address this point.

Response: We greatly appreciate your comment. As the reviewer pointed out, whether the anticancer effect of 2OCAA substances appears specifically in colorectal cancer has implications for the development of new drugs in the future. In fact, in the process of planning this experiment, an experiment was considered to check the effect on normal cells, but as described above, there is a limit to the amount of 2OCAA extracted from lichen, so there is a limit to using only one type. cancer cells. If additional materials are acquired in the future, molecular biology experiments will be conducted in various fields pointed out by the reviewer.

Comment 5: Since the authors report the occurrence of both early and/or late apoptosis after 2OCAA, the study itself lacks true mechanistic insights as to how 2OCAA elicits anti-tumor effects in human colorectal HCT116 cancer cells. In other words, crucial regulators of apoptotic machinery (cleaved PARP, caspase-3, caspase-9, and caspase-8) should be investigated to justify the obtained results also related to the control of mitochondrial permeability (e.g induction of Bax, inhibition of BCL-2).

Response: We greatly appreciate your comment. We also have thought it is necessary to elucidate the true apoptotic mechanism of 2OCAA. If additional materials are secured in the future, molecular experiments will be conducted on the specific mechanistic insights of apoptosis pointed out by the judges. We appreciate your understanding of the limitation of the experiment.

Comment 6: The manuscript must be revised by a native English speaker as it contains a lot of grammar errors and many sentences are very difficult to be followed.

Response: As the reviewer pointed out, our manuscript was revised by a native English speaker.  

Comment 7: The figures' resolution is very poor, and the figure legends should be more descriptive so that they stand alone from the text.

Response: Thank you for your suggestion. As the reviewer pointed out, we think that the quality of Figures, especially Figure 2C, is a little bad. In our laboratory, the resolution capability of our laboratory equipment, especially the microscope, is not very good. We would appreciate it if you understand this technical limitation to interpret the result. However, the figure efficiently showed the morphological characteristics of apoptotic cells exposed to 2OCAA. In addition, the figure legends were more descriptive by adding more information.

Comment 8: The discussion needs to be rewritten in order to be more fluid and not just a list of data.

Response: Thank you for your suggestion. As the reviewer pointed out, we rewrote the discussion in the Result and Discussion section (line 168-181).

Comment 9: Line 288, page 10 – Please indicate the number of collected events for flow cytometry analysis.

Response: Thank you for your suggestion. The number of collected events for flow cytometry analysis was indicated in Materials and Methods section (line 300).

Comment 10: Line 298, page 10 – Please indicate the thermal cycle protocol for qRT-PCR analysis (denaturation, annealing, extension….).

Response: Thank you for your suggestion. The thermal cycle protocol for qRT-PCR analysis was indicated in Materials and Methods section (Line 310-312).

Round 2

Reviewer 1 Report

The article " Anticancer activity of 2-O.caffeoyl alphitolic acid extracted from the lichen, Usnea barbata 2017-KL-10” has been improved but I still have some concerns:

  • Discussion of results has been improved but it is stil poor. Authors have improved apoptosis section but discussion in other sections is still poor and should be improved.

Author Response

The article " Anticancer activity of 2-O.caffeoyl alphitolic acid extracted from the lichen, Usnea barbata 2017-KL-10” has been improved but I still have some concerns:

Response: We appreciate the time and efforts of the reviewer in consideration of the original manuscript and believe that the comments have helped us to improve the manuscript. All the corresponding changes in response to the reviewer’s comments are marked in red.

Comment: Discussion of results has been improved but it is still poor. Authors have improved apoptosis section but discussion in other sections is still poor and should be improved.

Response: First, we greatly appreciate your comments, critical to improve overall quality of this manuscript. As recommended by the reviewer, we provided more information about the Discussion in the Result section (line 138-140).

Reviewer 2 Report

Reviewing is a time-intensive process – writing a review report can be almost as much work as writing a manuscript!  The authors do not report my suggestions although they claimed to have done it. This is not correct.

  • First, please, in Fig.2 B, replace cell viability (%) with metabolic activity (%)
  • In the abstract section line 28, please replace BCL-2 with Mcl-1 as stated by the authors. This is not in line with Fig.4
  • Why did the authors perform a student t-test instead of ANOVA for their statistical analysis? In general, the Student's t-test is used to compare the means between two groups, whereas ANOVA is used to compare the means among three or more groups. Please revise

Author Response

Comments and Suggestions for Authors

Reviewing is a time-intensive process – writing a review report can be almost as much work as writing a manuscript!  The authors do not report my suggestions although they claimed to have done it. This is not correct.

Response: We appreciate the time and efforts of the reviewer in consideration of the original manuscript to improve the manuscript. All changes in the text (additions and deletions over the previous submission) are marked in red.

Comment 1: First, please, in Fig.2 B, replace cell viability (%) with metabolic activity (%)

Response: First, we greatly appreciate your comments, critical to improve overall quality of this manuscript. As reviewer’s suggestion, we replaced cell viability (%) with metabolic activity (%) in Figure 2B.

Comment 2: In the abstract section line 28, please replace BCL-2 with Mcl-1 as stated by the authors. This is not in line with Fig.4

Response: We greatly appreciate your comment. We corrected it in the abstract section.

Comment 3: Why did the authors perform a student t-test instead of ANOVA for their statistical analysis? In general, the Student's t-test is used to compare the means between two groups, whereas ANOVA is used to compare the means among three or more groups. Please revise

Response: We greatly appreciate your comment. In the first revision, data were analyzed using one-way ANOVA with Tukey's post-hoc test. However, it should have been described in the Materials and Methods section and in the figure legend. We are really sorry about it. Edited in the manuscript (line 340 -343).